# IRF8 Regulates Intrinsic Ferroptosis through Repressing p53 Expression to Maintain Tumor Cell Sensitivity to Cytotoxic T Lymphocytes

**DOI:** 10.3390/cells12020310

**Published:** 2023-01-13

**Authors:** Dakota B. Poschel, Mercy Kehinde-Ige, John D. Klement, Dafeng Yang, Alyssa D. Merting, Natasha M. Savage, Huidong Shi, Kebin Liu

**Affiliations:** 1Department of Biochemistry and Molecular Biology, Medical College of Georgia, Augusta, GA 30912, USA; 2Georgia Cancer Center, Augusta, GA 30912, USA; 3Charlie Norwood VA Medical Center, Augusta, GA 30904, USA; 4Department of Pathology, Medical College of Georgia, Augusta, GA 30912, USA

**Keywords:** ferroptosis, IRF8, p53, tumor cell death, cancer immunotherapy, lipid nanoparticle gene therapy

## Abstract

Ferroptosis has emerged as a cytotoxic T lymphocyte (CTL)-induced tumor cell death pathway. The regulation of tumor cell sensitivity to ferroptosis is incompletely understood. Here, we report that interferon regulatory factor 8 (IRF8) functions as a regulator of tumor cell intrinsic ferroptosis. Genome-wide gene expression profiling identified the ferroptosis pathway as an IRF8-regulated pathway in tumor cells. IRF8.KO tumor cells acquire resistance to intrinsic ferroptosis induction and IRF8-deficient tumor cells also exhibit decreased ferroptosis in response to tumor-specific CTLs. *Irf8* deletion increased p53 expression in tumor cells and knocking out p53 in IRF8.KO tumor cells restored tumor cell sensitivity to intrinsic ferroptosis induction. Furthermore, IRF8.KO tumor cells grew significantly faster than WT tumor cells in immune-competent mice. To restore IRF8 expression in tumor cells, we designed and synthesized codon usage-optimized IRF8-encoding DNA to generate IRF8-encoding plasmid NTC9385R-mIRF8. Restoring IRF8 expression via a lipid nanoparticle-encapsulated NTC9385R-mIRF8 plasmid therapy suppressed established tumor growth in vivo. In human cancer patients, nivolumab responders have a significantly higher IRF8 expression level in their tumor cells as compared to the non-responders. Our data determine that IRF8 represses p53 expression to maintain tumor cell sensitivity to intrinsic ferroptosis.

## 1. Introduction

Breakthroughs in immune checkpoint inhibitor (ICI) immunotherapy have resulted in durable efficacy in many types of human cancers. However, even for responsive cancers, such as melanoma and non-small cell lung carcinoma, there are still a large portion of patients who do not respond to ICI immunotherapy [1]. In certain types of human cancers, including colorectal cancer and pancreatic cancer, the vast majority of patients do not respond to ICI immunotherapy. It is known that cytotoxic T lymphocytes (CTLs) infiltrate the tumor in these non-responder cancers [2,3]. Therefore, compensatory immune suppressive mechanisms, such as osteopenia (OPN), CD47 and CD73, may contribute, at least in part, to the non-response to ICI immunotherapy in these non-responsive cancers [4,5,6,7,8]. ICI immunotherapy acts through blocking immune suppression to activate CTLs. The activated CTLs then suppress tumors by killing the tumor cells using various effector mechanisms, including the Fas-FasL and the granzyme B and perforin pathways. Therefore, in addition to the activation and effector function of tumor-specific CTLs, the target tumor cells must also be sensitive to the induction of cell death [9]. Resistance to cell death is a hallmark of human cancer [10,11,12,13,14]. This dysregulation of the cell death pathway is regulated by various mechanisms in tumor cells [14,15,16,17,18]. Elucidating the molecular mechanisms underlying tumor cell death pathway dysregulation and developing mechanism-based therapies to sensitize tumor cell to death induction are, therefore, potentially effective approaches to enhance cancer immunotherapy efficacy.

Ferroptosis is a recently discovered lipid reactive oxygen species (ROS) and iron-dependent regulated cell death pathway [19,20]. It was first discovered from the phenomenon that a compound eradicator of RAS and ST-expressing cells (erastin) can selectively induce RAS-expressing tumor cell death in a manner that is different from apoptosis and other known cell death pathways [19,21]. Erastin inhibits the amino acid antiporter Xc- that is highly present in the phospholipid bilayers of mammalian cells. System Xc- is a heterodimer and consists of the solute carrier family 7 member 11 (SLC7A11) and solute carrier family 3 member 2 (SLC3A2) subunits. The main function of system Xc- is to mediate changes of cystine and glutamate in and out of a cell. The cystine, once taken into the cell, is reduced to cysteine for the synthesis of glutathione (GSH). GSH then reduces ROS through the activity of glutathione peroxidases (GPX4) [22]. Erastin inhibits system Xc- to cause decreased GSH levels and decreased GPX4 activity, leading to lipid ROS accumulation and ferroptosis [22]. GPX4 converts GSH into oxidized glutathione (GSSG) and reduces the cytotoxic lipid peroxides (L-OOH) to alcohols (L-OH) [19]. Inhibition of GPX4 activity leads to the accumulation of lipid peroxides, resulting in ferroptosis [22]. Ras Selective Lethal-3 (RSL3) is another ferroptosis inducer that directly binds to GPX4 to inhibit its activity, resulting in ROS accumulation and ferroptosis [23,24]. It was also recently determined that CTLs secrete IFNγ to activate the STAT1-IRF1 axis, in order to repress the expression of SLC7A11 and SLC3A2 to induce the extrinsic ferroptosis pathway in tumor cells [25]. Therefore, in addition to the Fas-FasL and granzyme B-perforin pathways, ferroptosis is another CTL-mediated cell death pathway that may also contributes to the CTL effector function in ICI immunotherapy [25]. p53 is a key regulator of ferroptosis and can either promote or suppress ferroptosis by cellular context-dependent mechanisms [26,27,28,29,30,31,32,33]. How p53 is regulated in the ferroptosis pathway is currently incompletely understood.

Interferon regulatory factor 8 (IRF8), formerly termed interferon consensus sequence-binding protein (ICSBP), was originally identified as an essential lineage-specific transcription factor for myeloid cell differentiation [34]. The function of IRF8 has since been extended to the regulation of myeloid, B, and T cells [35,36,37,38,39,40,41,42]. Furthermore, it has also been expanded to include non-hematopoietic cells [36,43,44,45,46,47,48,49,50,51]. IRF8 functions as a tumor suppressor [48,50,52,53], its function, at least in part, depending on its function in the regulation of tumor cell apoptosis [51,54,55]. However, whether IRF8 regulates other cell death pathways in tumor cells is unknown. Here, we report that IRF8 functions as a ferroptosis regulator that represses p53 expression to maintain tumor cell sensitivity to intrinsic ferroptosis induction by tumor-specific CTLs. The IRF8-p53 axis thus represents a previously uncharacterized ferroptosis pathway in tumor cells.

## 2. Materials and Methods

### 2.1. Mice and Tumor Cell Line

IRF8 knock-out (KO) mice were created as described previously and were provided by Dr. Keiko Ozato (National Institutes of Health, Bethesda, MD, USA). The IRF8 KO mouse colony is maintained in the Augusta University laboratory animal facility. All IRF8 KO mice were genotyped as previously described [34]. C57BL/6 and BALB/c mice were obtained from the Jackson Laboratory (Bar Harbor, ME, USA). All studies with mice were approved by Augusta University Institute Animal Care and Use Committee (protocol # 2008-0162). Murine colon tumor cell line CT26 was obtained from American Type Culture Collection (ATCC, Manassas, VA, USA). ATCC authenticates cell lines by morphology, immunology, DNA fingerprint, and cytogenetics. The CMS4 cell line was provided by Dr. A. DeLeo (University of Pittsburgh, Pittsburgh, PA) [56]. All cell lines were tested bi-monthly for mycoplasma and were mycoplasma-free at the time of the experiments.

### 2.2. Generation of IRF8.KO Tumor Cell Lines

WT and IRF8 KO mice were subcutaneously injected with methylcholanthrene (MCA, Sigma-Aldrich, St Louis, MO, USA) at 100 µg in 100 µL of peanut oil per mouse, as previously described [57]. Tumors were dissected from the mice approximately 3 months later and digested with collagenase solution [Collagenase, Sigma-Aldrich cat#C0130: 1 mg/mL, Hyaluronidase, Sigma-Aldrich, Cat#H3506: 0.1 mg/mL, and DNase I: 30 U/mL). The tumor cell mixture was then cultured for at least 10 passages to establish stable tumor cell lines, as previously described [58]. Two WT (MC654 and MC659) and two IRF8 KO tumor cell lines (MC010 and MC011) were created. We then isolated genomic DNA from these four tumor cell lines and performed PCR-based genotyping using the IRF8 KO mouse genotyping procedures [34]. The PCR primers are: Forward: 5′-CATGGCACTGGCCAGATGTCTTCC-3′, WT reverse: 5′-CTTCCAGGGGATACGGAACATGGTC-3′, and KO reverse: 5′-CGAAGGAGCAAAGCTGCTATTGGCC-3′.

### 2.3. Generation of p53 KO Tumor Cell Lines

p53 KO tumor cell lines were created using the pLentiCRISPR V2 system [59] and the procedures as described [7]. Briefly, HEK293FT cells were transfected with pCMV-VSV-G (Addgene Cat #8454), psPAX2 (Addgene, Cat#12260), and lentiCRISPRv2 (Genscript, Piscataway, NJ, USA) plasmids containing scramble (GGAAGACTTAGTCGAATGAT), two *Trp53*-specific (sgRNA1: AGTGAAGCCCTCCGAGTGTC; sgRNA2: AACAGATCGTCCATGCAGTG) sgRNA-encoding sequences, respectively, using lipofectamine 2000 (Invitrogen, Carlsbad, CA, USA). After 72 h, culture supernatant/lentiviral particles were harvested and used to transduce MC010 and MC011 cells. The tumor cells were cultured for 24 h and selected by adding puromycin to the culture medium. Cell phenotype was confirmed using a Western blot analysis.

### 2.4. Histology and Immunohistochemical Analysis

Cell blocks were prepared by centrifugation. The cell blocks and tumor tissues were fixed in 10% formalin, paraffin-embedded, sectioned and stained with hematoxylin and eosin at the Electron Microscopy and Histology core facility in Augusta University. The tissue sections were then stained with anti-mouse Ki67 antibody (Abcam, Cat# ab15580) at the Georgia Research Pathology Core Lab. The cell blocks were also stained with anti-vimentin (Santa Cruz, Cat#sc-6260) at the Electron Microscopy and Histology Core Facility. IHCs were performed, according to the previously described procedures [60].

### 2.5. RNA Sequencing Analysis

Total RNA was purified from the tumor cells using the Direct-zol RNA Microprep Kit, according to the manufacturer’s instructions (Zymo Research, Irvine, CA, USA). RNA sequencing was performed by MedGenome Inc. (Foster City, CA, USA). Reference genome (mm10) and gene model annotation files from Ensembl were used for aligning reads using STARv2.5. The reads were quantified using HTSeqv0.6.1. A differential gene expression analysis was performed using DESeq2 v1.36. Ranked-log2 fold changes from the DESeq2 analysis were used as inputs for the analysis [61]. GO pathway enrichment was performed with clusterProfiler V4.0. A heatmap was generated using ComplexHeatmap v2.12.0. A ferroptosis pathway map was generated in Pathview v1.36 based on KEGG collections of the mouse ferroptosis pathway gene set. The entire dataset is deposited in GEO (accession # GSE213856). 

### 2.6. Ferroptosis Induction and Quantification

The tumor cells were cultured in a 24-well plate at 1 × 10^5^ cells/well in 1 mL of RPMI medium plus 10% FBS. The cells were then treated with Erastin (SelleckChem, Houston, TX, USA) and RSL3 (R&D Systems, Minneapolis, MN, USA), respectively, for approximately 24 h. Both floating and adherent cells were collected, stained with propidium iodide (PI, Sigma-Aldrich, 0.5 μg/100 μL), and analyzed with a FACSCaliber flow cytometer (BD Biosciences, San Diego, CA, USA). Cell death was quantified as the percentage of PI+ cells using the FlowJo software.

### 2.7. Tumor Cell and Tumor-Specific CTL Co-Culture

2/20 CTLs were maintained by weekly stimulation with AH1 peptide and irradiated spleen cells as feeder cells, as previously described [62]. The tumor cells were seeded 1 × 10^5^/well into a 96-well U-bottom plate. 2/20 CTLs were purified using lymphocyte separation medium (Corning, NY, USA) added at a 2:1 (tumor:effector) ratio and cultured overnight. Ferrostatin-1 (Sigma-Aldrich, 10 μM) or solvent was added to each well. Both floating and adherent cells were harvested, stained with CD3-specific monoclonal antibody (BioLegend, San Diego, CA, USA) and PI, and analyzed with the FACSCalibur flow cytometer. CD3^−^ tumor cells were gated for PI^+^ cells. 

### 2.8. Flow Cytometry

Tumor tissues were digested with collagenase solution. The digested tumor cells were stained with Zombie UV (BioLegend) followed by surface antibodies, including IgG-FITC and CD3-APC (BioLegend), and analyzed on a LSRFortessa flow cytometer with BD Diva 8.01 (BD Biosciences). The cultured tumor cells were analyzed on a FACSCalibur with CellQuest Pro (BD Biosciences). All flow cytometry data were analyzed with FlowJo v10.6.0 (BD Biosciences).

### 2.9. Lipid ROS Analysis

The tumor cells were cultured overnight at 1.5 × 10^5^ cells/well in 24 well plates. After this, they were treated for 1 h with 3µM of RSL-3 or equivalent volume vehicle (DMSO), and then stained with Image-IT BODIPY-C11 (Invitrogen) for 30 min at 37 °C. Floating and adherent cells were harvested and flow cytometry was run immediately using a FACSCalibur. Lipid ROS was determined by the ratio of the mean fluorescent intensity in FL1 (530/30) divided by the mean fluorescent intensity in FL2 (585/42).

### 2.10. GSH Measurement

The cells were seeded at 10,000 cells/well in a 96-well plate overnight. They were then treated with 3 µM of RSL-3 or equivalent volume vehicle (DMSO) for 30 min. The GSH was measured using the GSH-Glo Kit (Promega), according to the manufacturer’s instructions. Luminescence was determined using a Cytation5 plate reader (Biotek).

### 2.11. Cationic Lipid Nanoparticle

Cationic lipid DOTAP(N-[1-(2,3-Dioleoyloxy)propyl]-N,N,N-trimethylammonium methyl-sulfate)-Cholesterol (1:1 molar ratio) was manufactured at T&T Scientific Corp (Knoxville, TN, USA). Codon usage-optimized mouse IRF8-encoding DNA was designed and synthesized in Genscript Inc (Piscataway, NJ, USA) (Appendix A). The synthesized IRF8 DNA was cloned to the non-immunogenic nanoplasmid NTC9385R in Nature Technology Corp (Lincoln, NE, USA) to generate codon usage-optimized mouse IRF8 expressing plasmid (NTC9385R-mIRF8). Large-scale NTC9385R-mIRF8 plasmid production was carried out by fermentation in Nature Technology. To formulate lipid nanoparticle-encapsulated DNA, DOTAP-Chol and NTC9385R-mIRF8 were diluted in 5% Dextrose (Cat # 76313-652, VWR International), respectively. The diluted DOTAP-Cholesterol (10 mM) and NTC9385R-mIRF8 plasmid DNA (0.5 mg/mL) were then mixed in a 1:1 ratio and incubated at room temperature for 30 min to produce lipid nanoparticle-encapsulated NTC9385R-mIRF8 plasmid DNA (mIRF8-LNP).

### 2.12. mIRF8-LNP Therapy

CT26 cells (2 × 10^5^ cells/mouse) were subcutaneously injected into BALB/c mice. The tumor-bearing mice were treated with Vector-LNP and the mIRF8-LNP, respectively. They were injected with 100 μL of nanoparticle each time via intravenous injection. 

### 2.13. Western Blotting Analysis

The cells were lysed in total lysis buffer (20 mM of HEPES, pH 7.4, 20 mM of NaCl, 10% glycerol, 1% TritonX100, Proteinase inhibitor cocktails) for 60 min on ice, then centrifuged to removed debris. Cell lysates were separated in a 4–20% gradient denature SDS-polyacrylamide gel (Biorad, Hercules, CA, USA). Membrane was blocked for 1 h at room temperature in 5% milk in PBST, then incubated with primary antibodies (anti-p53: Cell Signaling Tech cat# 2524; anti-GPX4: Cell Signaling Tech Cat#52455) overnight at 4 °C, followed by 2nd antibody and detection using the ECL system, as previously described [28]. The membranes were stripped and re-probed with anti-β-actin antibody (Sigma-Aldrich, Cat# 9026). The original Western blots are presented in the supplemental material (Figure S2). 

### 2.14. Single-Cell RNA Sequencing Dataset Analysis

Single-cell RNA sequencing datasets of human melanoma patients were extracted from the GEO database (GSE115978) [63]. The cells were annotated according to dataset designation and subdivided by cell type and pre-/post nivolumab immunotherapy using OmniBrowser.

### 2.15. Statistical Analysis

All statistical analysis was performed using Graphpad Prism 9 (San Diego, CA, USA). The *p* value was determined using a two-tailed Student’s t-test.

## 3. Results

### 3.1. Generation and Characterization of IRF8 KO Tumor Cell Lines

To establish IRF8-deficent tumor cells lines, we treated IRF8 knock-out mice and wild-type (WT) C57BL/6 mice with the carcinogen methylcholanthrene (MCA). Tumors from WT and IRF8.KO mice were resected and digested with collagenase solution to form a single cell suspension. The single cells were cultured to establish stable tumor cell lines. Two WT (MC654 and MC659) and two IRF8.KO (MC010 and MC011) cell lines were established as stable tumor cell lines. Genotyping validated that MC654 and MC659 are IRF8 WT cell lines and MC010 and MC011 are IRF8 KO cell lines (Figure 1A). An analysis of the tumor cell lines by a board-certified pathologist determined that all cell lines are undifferentiated high-grade fibrosarcoma (Figure 1B). An analysis of the tumor cells for proliferation determined that these tumor cell lines are highly proliferating cells (Figure 1C). We also stained for vimentin, a marker of mesenchymal cells or cells that have undergone mesenchymal transition [64]. As expected, although the tumor cells are heterogenous in vimentin level, almost all cells express vimentin protein (Figure 1D).

### 3.2. IRF8 Deficiency Leads to Dysregulated Expression of Genes in the Ferroptosis Pathway

IRF8 is a transcription factor [65]. To determine IRF8-regulated genes in the tumor cells, RNA was isolated from the cultured tumor cells in vitro and analyzed using RNA-Seq. Two pairs of WT and IRF8 KO cells were used. An analysis of differentially expressed genes between these two pairs of WT and IRF8 KO cell lines identified the ferroptosis pathway as one of the enriched pathways (Figure 2A,B). The expression of a set of ferroptosis regulatory genes was regulated by IRF8 (Figure 2C,D). p53 is a known ferroptosis regulator with both promoting and suppressive functions in ferroptosis, depending on the cellular context [66]. We observed that knocking out IRF8 increased p53 expression level in tumor cells (Figure 2C,D). A Western blotting analysis validated that the p53 protein level is increased in IRF8.KO tumor cell lines E and Appendix A).

### 3.3. IRF8 Regulates Intrinsic Ferroptosis through Repressing p53 Expression

We next sought to determine whether IRF8 regulates ferroptosis. We used erastin and RSL3 as ferroptosis inducers [19,21,67]. Erastin inhibits system Xc-, thereby preventing the cellular import of cysteine, to induce extrinsic ferroptosis [68], and RSL3 is a GPX4-specific inhibitor that induces intrinsic ferroptosis [23]. The WT and IRF8.KO tumor cells were treated with erastin and RSL3, respectively, and analyzed for cell death. Both the WT and IRF8.KO tumor cells were not sensitive to erastin (Figure 3A,B). The WT tumor cells were sensitive to the RSL3 induction of cell death. However, knocking out IRF8 significantly decreased the tumor cell sensitivity to ferroptosis induction by RSL3 (Figure 3A,B). This finding indicates that IRF8 regulates intrinsic ferroptosis.

P53 has been shown to either promote or suppress ferroptosis [27,31,32]. P53 is upregulated in IRF8 KO tumor cells and is undetectable in the WT tumor cells (Figure 2E). To determine whether IRF8 regulates ferroptosis through p53, we knocked out Trp53 in the IRF8.KO tumor cell lines (Figure 3C and Appendix A) and analyzed tumor cell sensitivity to ferroptosis induction. Knocking out Trp53 significantly increased IRF8.KO tumor cell sensitivity to ferroptosis induction by RSL3 in both IRF8.KO tumor cell lines in vitro (Figure 3D,E). These findings indicate that p53 functions as a ferroptosis suppressor and IRF8 represses p53 to maintain tumor cell sensitivity to intrinsic ferroptosis.

GPX4 converts GSH to GSSG and RSL3 is a GPX4-selective inhibitor that binds to GPX4 to inhibit its enzymatic activity [19]. The Western blotting analysis revealed that RSL3 does not significantly change the GPX4 protein level in the tumor cells (Appendix A). However, an analysis of the GSH levels revealed that RSL3 treatment did not cause a significant accumulation of GSH in the tumor cells (Appendix A). Since lipid ROS production is suppressed by GPX4, we observed, as expected, that RSL3 treatment significantly increased the lipid ROS level in the tumor cells. Consistent with sensitivity to ferroptosis induction by RSL3, the WT tumor cell lines exhibit a higher lipid ROS level than the IRF8 KO tumor cell lines (Appendix A). 

### 3.4. IRF8 Controls Tumor Cell Sensitivity to Tumor-Specific CTL-Induced Ferroptosis

CTLs kill tumor cells through inducing tumor cell ferroptosis in ICI immunotherapy [25]. To determine whether tumor cell-expressed IRF8 regulates CTL-induced ferroptosis, we used the tumor cell line CMS4 and the CMS4 tumor cell-specific 2/20 CTL line. The 2/20 CTL line is a H2Ld-restricted CTL line that recognizes the AH1 peptide of a viral protein gp70 that is expressed in CMS4 tumor cells [62]. We then used the CMS4 tumor cell lines stably expressing an IRF8 dominant-negative mutant (CMS4.K79E). The IRF8K79E mutant was functionally characterized previously and determined to function as a dominant-negative mutant [55,69]. As expected, loss of IRF8 function in the CMS4.K79E cell line significantly decreased tumor cell sensitivity to RSL3-induced cell death (Figure 4A). To determine whether CTLs kill tumor cells through inducing tumor cell ferroptosis, the tumor cells and 2/20 CTLs were then co-cultured in vitro and analyzed for tumor cell death. The CD3^−^ tumor cells were gated (Figure 4C) and quantified for cell death (Figure 4D). IRF8 mutant CMS4.K79E cells exhibited decreased sensitivity to 2/20 CTL-induced cell killing as compared to the vector control cells (Figure 4D), indicating that IRF8 regulates 2/20 CTL-induced tumor cell death. To determine whether this IRF8-regulated and CTL-induced target tumor cell death is through ferroptosis, ferroptosis inhibitor ferrostatin-1 was added to the tumor cell-2/20 CTL co-culture. Again, the CD3^−^ tumor cells were gated (Figure 4C) and quantified for cell death (Figure 4D). The inhibition of ferroptosis significantly decreased CTL-induced control tumor cell death, but not mutant IRF8-expressing CMS4.K79E tumor cell death (Figure 4D). Taken together, our observations indicate that IRF8 regulates target tumor cell sensitivity to ferroptosis induction by tumor-specific CTLs.

### 3.5. Loss of IRF8 Expression Results in Increased Tumor Growth In Vivo

To determine whether the above findings can be translated to tumor growth control in vivo, WT and IRF8.KO tumor cells were injected into immune-competent syngeneic mice, and tumor growth was monitored. At the tumor cell dose analyzed, the WT MC654 tumors grew initially and then disappeared. In contrast, the MC011 IRF8.KO tumor grew constantly and rapidly in the immune-competent mice (Figure 5).

### 3.6. IRF8-Encoding Plasmid DNA Nanoparticle Therapy Suppresses Tumor Growth In Vitro

The above findings determined that IRF8 represses p53 to maintain tumor cell sensitivity to ferroptosis, and the loss of IRF8 expression or function decreases tumor cell sensitivity to CTL-induced ferroptosis and promotes tumor growth in vivo. IRF8 is often silenced in tumor cells by DNA hypermethylation at its promoter [51,55,70,71]. CT26 is a tumor cell line with silenced IRF8 expression [55]. We therefore hypothesized that restoring IRF8 expression to tumor cells should sensitize the tumor cell to ferroptosis to suppress tumor growth in vivo. To test this hypothesis, we made use of an IRF8-encoding plasmid nanoparticle. We first synthesized codon usage-optimized IRF8-coding DNA (Figure 6A and Appendix A) and cloned this DNA to the non-immunogenic nanoplasmid NTC9385R [72] to generate NTC9385R-mIRF8 plasmid. DOTAP-cholesterol is a cationic lipid nanoparticle that has been shown to selectively deliver DNA to tumors [13,73,74,75]. We encapsulated NTC9385R-mIRF8 to DOTAP-Cholesterol to formulate a DNA nanoparticle (Figure 6A) and treated CT26 tumor-bearing mice with this DNA nanoparticle (Figure 6B). NTC9385R-mIRF8 plasmid DNA nanoparticle therapy suppressed the established tumor growth (Figure 6C). At the endpoint, we used the viability stain Zombie UV to assess the viability of the tumor cells, and found a significantly lower viability in the treatment group in vivo (Figure 6D). These findings indicate that restoring IRF8 expression via a lipid nanoparticle therapy is effective in the suppression of tumor growth in vivo by causing increased tumor cell death.

### 3.7. IRF8 Expression Level in Tumor Cells Correlates Patient Response to ICI Immunotherapy

To determine the translational significance, we analyzed human melanoma scRNA-seq datasets from responder and non-responder patients at the single cell level. As expected, IRF8 is highly expressed in immune cells (Figure 7A). IRF8 is also expressed in tumor cells (Figure 7A,B). A comparison of IRF8 expression level in tumor cells between patients who responded to nivolumab and patients who did not determined that the IRF8 expression level is significantly higher in the responders than in the non-responders (Figure 7C). These findings indicate that tumor cell IRF8 expression positively correlates with melanoma patient response to ICI immunotherapy. 

## 4. Discussion

Previous studies have identified ferroptosis as a CTL-induced tumor cell death pathway [25]. In this cell death pathway, activated CTLs secrete IFNγ to activate STAT1. STAT1 then upregulates IRF1 to repress the expression of SLC7A11 and SLC3A2, the two subunits of the glutamate-cystine antiporter system xc-, to limit tumor cell cystine uptake, resulting in tumor cell lipid peroxidation and death [25]. IRF1 thus bridges CTLs to ferroptosis. In this study, we identified IRF8 as a ferroptosis regulator in tumor cells. Like IRF1, IRF8 is a member of the IRF transcription factor family. However, unlike IRF1, IRF8 deletion did not change the expression of SLC3A2 or SLC7A11 in tumor cells. Instead, IRF8 represses p53 expression to regulate the intrinsic ferroptosis. We therefore identified IRF8 as a previously uncharacterized ferroptosis regulator in tumor cells and determined that IRF8 represses p53 expression to maintain tumor cell sensitivity to CTL lyric activity. Like IRF1, IRF8 is also inducible by IFNγ [76]. It is, therefore, likely that IRF8 is upregulated in tumor cells by activated CTLs in the tumor microenvironment. However, the function of the CTL-IFNγ-activated IRF8 pathway in tumor cell ferroptosis requires further study.

p53 has dual and opposite functions in ferroptosis and its function in regulating ferroptosis is apparently cellular context-dependent [26,27,28,29,30,31,32,33]. It has been shown that p53 represses SLC7A11 to induce extrinsic ferroptosis, which releases arachidonate 12-Lipoxygenase, 12S type (ALOX12) from SLC7A11 inhibition [27]. However, an apparently contradictory role for p53 in ferroptosis has also been found. The stabilization of p53 leads to the expression of CDKN1A/p21, which can delay the onset of ferroptosis in the setting of cystine deprivation [32]. A high expression of p53 increases ferroptosis in multiple models and p21 suppresses ferroptosis [32]. p53 is also capable of suppressing the extrinsic ferroptosis pathway by blocking DPP4 activity [31]. In this study, we observed that p53 functions as a ferroptosis suppressor and IRF8 represses p53 expression to maintain tumor cell sensitivity to ferroptosis induction. As RSL3 directly inhibits GPX4 activity, and we observed that p53 suppresses RSL3-induced ferroptosis, our data thus indicate that p53 blocks the RSL3 inhibition of GPX4. We observed that increased p53 expression does not change tumor cell sensitivity to extrinsic ferroptosis induction by erastin, a selective inhibitor of system Xc^−^ [67]. Therefore, our data indicate that p53 regulates the intrinsic ferroptosis pathway. We also determined that RSL3 treatment significantly increases the lipid ROS level but does not decrease the GSH level. As RSL3 directly binds to GPX4 to inhibit its enzymatic activity [19], our findings thus indicate that the IRF8-p53 axis acts on targets between GPX4 and lipid ROS. The molecular targets of the IRF8-p53 axis remain to be determined and require further studies.

The function of IFN-regulated genes is essential for human cancer patient response to ICI immunotherapy [77]. IRF8 is an IFNγ-regulated gene [76]. A low level of IRF8 expression in tumors is associated with poor prognosis in human hepatocellular carcinoma (HCC), and gene set enrichment analysis identified the IFNγ and PD-1 signaling signatures as the top suppressed pathways in patients with IRF8-low HCC [52]. High IRF8 expression in tumor cells is correlated with a better response to immunotherapy and chemotherapy in human breast cancer [53]. In this study, we observed that IRF8 expression level is significantly higher in tumor cells at the single cell level in responders than in non-responders in response to nivolumab immunotherapy in human melanoma patients. Consistent with this phenomenon, lipid nanoparticle-mediated IRF8-encoding plasmid DNA therapy significantly increased tumor cell death and suppressed established tumor growth. It is possible that the IRF8-p53 ferroptosis pathway contributes, at least in part, to tumor cell destruction by the activated host CTLs. Therefore, IRF8-encoding plasmid DNA lipid nanoparticle therapy is a potentially effective approach to increase IRF8 expression in the target tumor cells to convert ICI immunotherapy non-responders to responders, which warrants further study.

## Figures and Tables

**Figure 1 cells-12-00310-f001:**
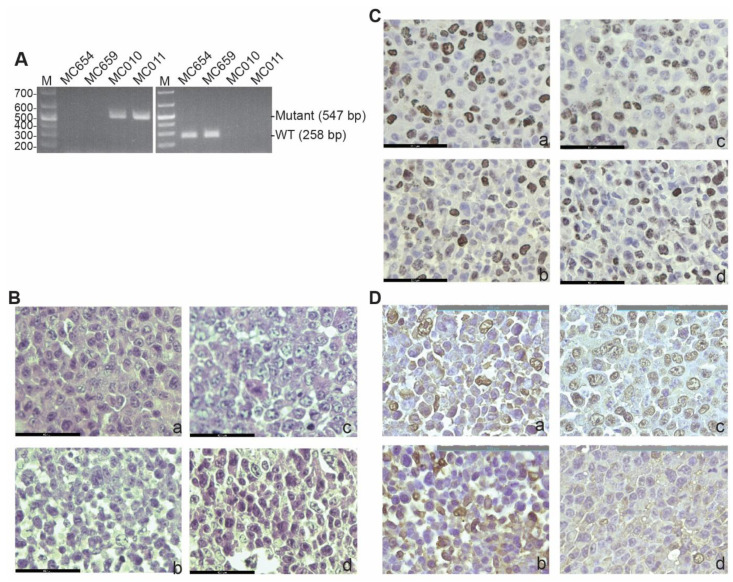
Generation of WT and IRF8 KO tumor cell lines. (**A**) Genomic DNA was isolated from the indicated tumor cell lines and used as a template for PCR-based genotyping with *Irf8* WT (**left panel**) and mutant-specific (**right panel**) primers, as described in the method. The DNA size markers are indicated to the left of each gel. (**B**) Tumor cell blocks were analyzed by H&E staining. Shown are the tumor cell morphologies. a: MC654, b: MC659, c: MC010, and d: MC011. The scale bar (62.1 μm) is in the bottom left. (**C**) Tumor cell block sections were analyzed by IHC using Ki67-specific antibody. The brown color indicates Ki67 staining. a–d: same as in (**B**). The scale bar (62.1 μm) is in the bottom left. (**D**) The tumor cell block sections were analyzed for vimentin by IHC using vimentin-specific antibody. The brown color indicates vimentin staining. a–d: same as in (**B**). The scale bar (100 μm) is in the top right.

**Figure 2 cells-12-00310-f002:**
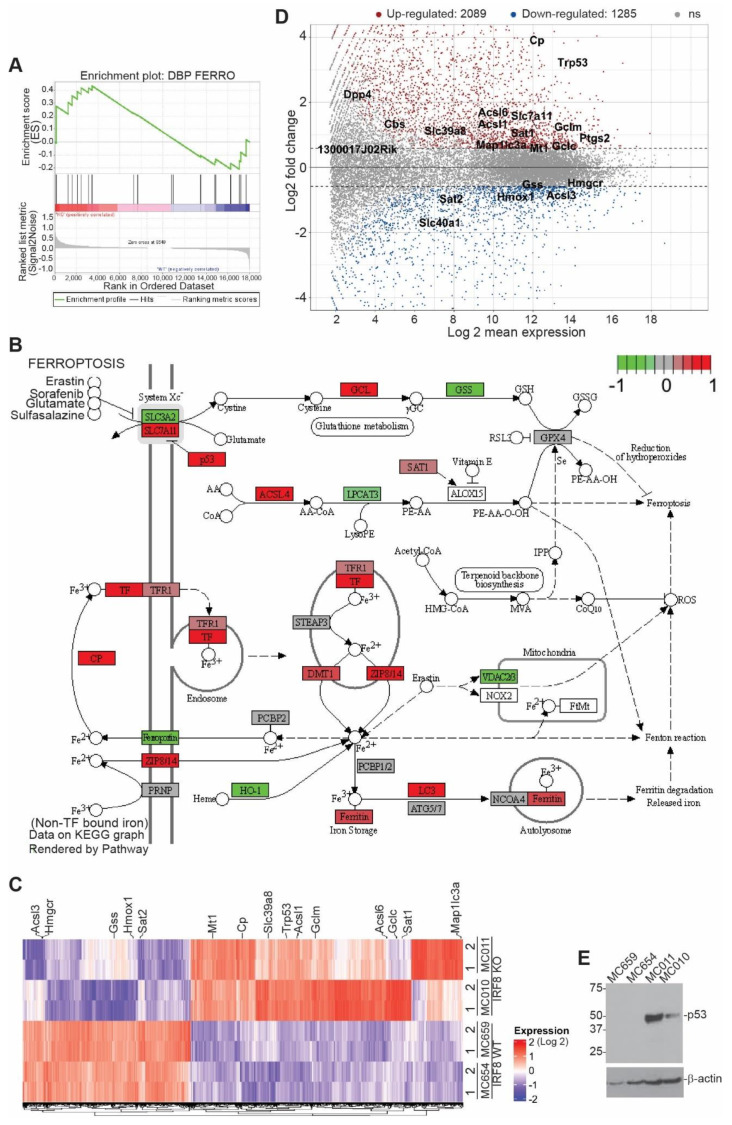
Deletion of *Irf8* gene alters the expression of ferroptosis pathway genes. (**A**). RNA sequencing analysis of gene expression of WT and IRF8 KO tumor cell lines. Shown is gene set enrichment analysis of ferroptosis pathway using ranked log2 fold changes. (**B**). KEGG Pathway view of the differentially expressed genes in murine ferroptosis pathway (adjusted *p*-value < 0.05). Relatively up-regulated genes (log2FC > 1) are shown in red, and relatively down-regulated genes (log2FC < 1) are shown in green. (**C**). RNA sequencing analysis of gene expression in the indicated pairs of IRF8 WT and KO tumor cell lines. Shown is heatmap of genes differentially expressed between WT andIRF8 KO tumor cells. Ferroptosis pathway genes are labelled on the right (1 and 2: replicates, absolute log2FC > 0.59, adjusted *p*-value < 0.05). (**D**). Ratio intensity (MA) plot of genes differentially expressed between WT and IRF8 KO tumor cells (adjusted *p*-value < 0.05). Ferroptosis pathway genes are labelled. Upregulated genes (log2FC > 0.59) are shown in red, and downregulated genes (log2FC < 0.59) are shown in blue above and below the dotted line respectively. (**E**). Western blotting analysis of the IRF8-regulated ferroptosis genes as indicated in WT and IRF8 KO tumor cells.

**Figure 3 cells-12-00310-f003:**
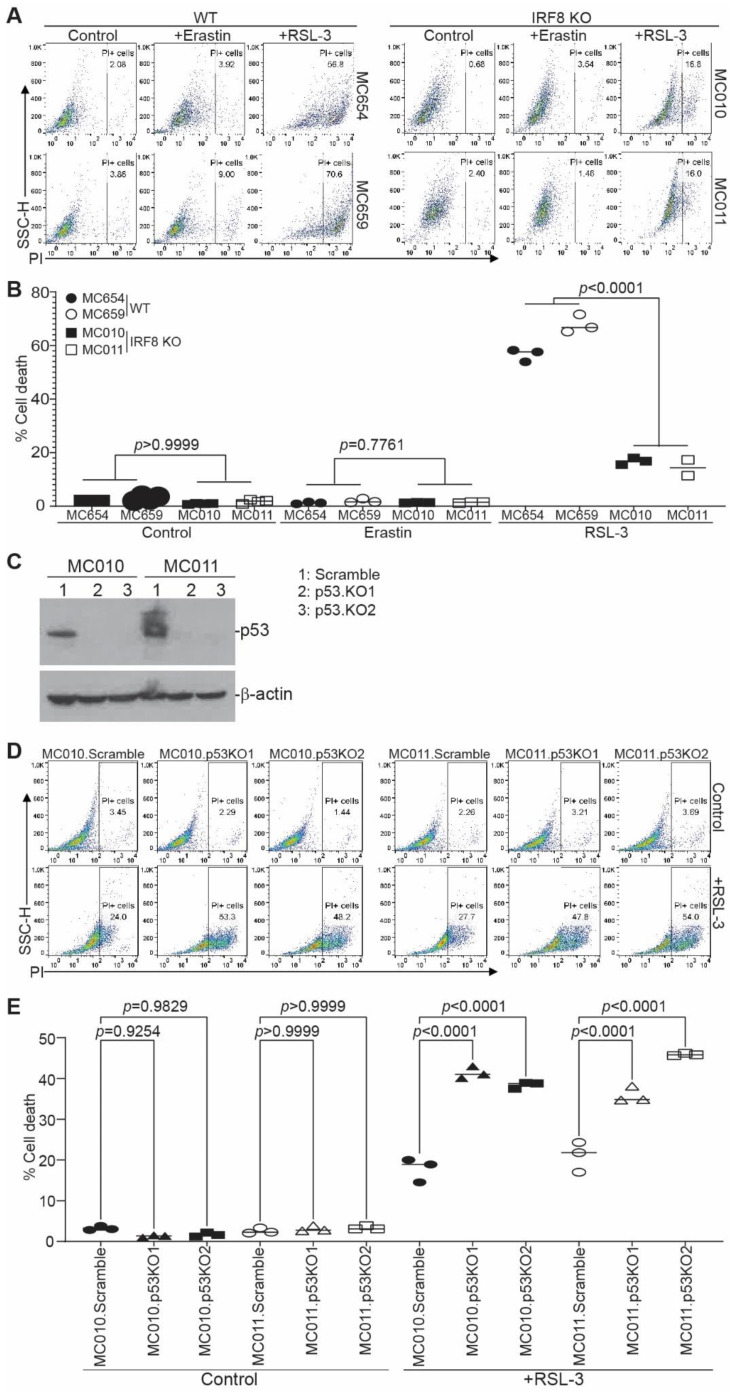
IRF8 regulates intrinsic ferroptosis in tumor cells. (**A**) Tumor cells were cultured in the presence of erastin (20 μM) and RSL-3 (0.1 μM) for approximately 24 h. Cells were collected, stained with PI, and analyzed by flow cytometry. Shown are representative gating strategies for wild-type and IRF8KO cells. (**B**) The % cell death (% PI^+^ cells) of the two WT cell lines (MC654 and MC659) and the two IRF8 KO cell lines (MC010 and MC011) were quantified, pooled, and presented. (**C**) Trp53 was knocked out in MC011 cells by CRISPR and analyzed for p53 protein level by Western blotting analysis. The lanes on the Western blot show the scramble and the 2 p53 KO cell lines of the 2 IRF8 KO cell lines (MC010 and MC011). (**D**) The WT and p53 KO cells were treated with RSL3 and analyzed for cell death as in (**A**). (**E**) Quantification of cell death as shown in (**D**).

**Figure 4 cells-12-00310-f004:**
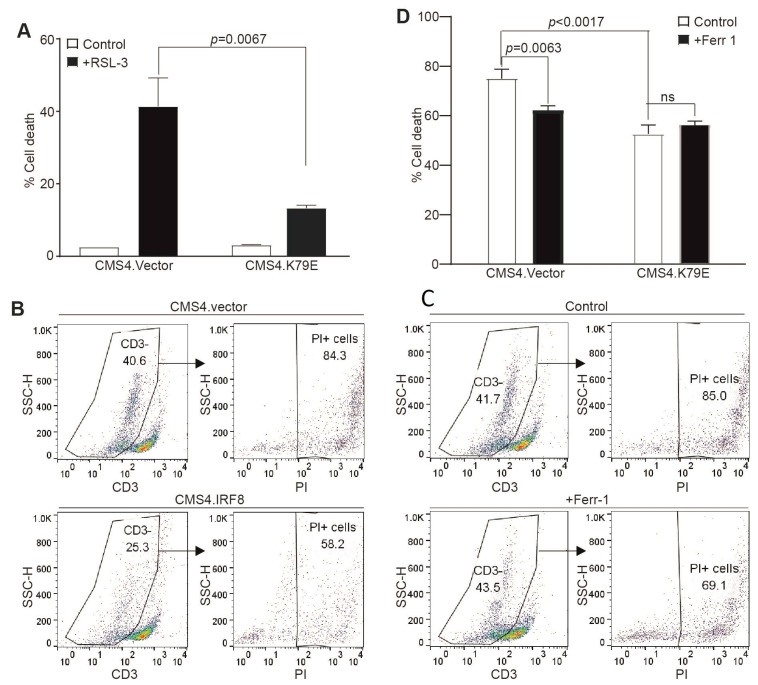
IRF8 regulates tumor cell sensitivity to ferroptosis and CTL lytic activity. (**A**) IRF8 WT vector control (CMS4.vector) and IRF8 mutant (CMS4.K79E) tumor cells were cultured in the presence of RSL-3 for approximately 24 h, stained with PI, and analyzed by flow cytometry to quantify cell death. (**B**) CMS4.vector and CMS4.K79E tumor cells were co-cultured with the tumor-specific CTLs at a 2:1 ratio for approximately 24 h. Cell mixture was collected, stained with CD3-specific monoclonal antibody and PI, and analyzed by flow cytometry. Shown are representative gating strategies. The CD3^−^ tumor cells were gated to quantify PI^+^ tumor cells. (**C**) CMS4.vector tumor cells were co-cultured with the tumor-specific 220 CTL (2:1 ratio) in the absence (control) or presence of ferrostatin (Ferr-1 1 μM) for approximately 24 h. Shown are representative gating strategies as in (**A**,**D**). The % cell death as shown in (**B**,**C**) was quantified.

**Figure 5 cells-12-00310-f005:**
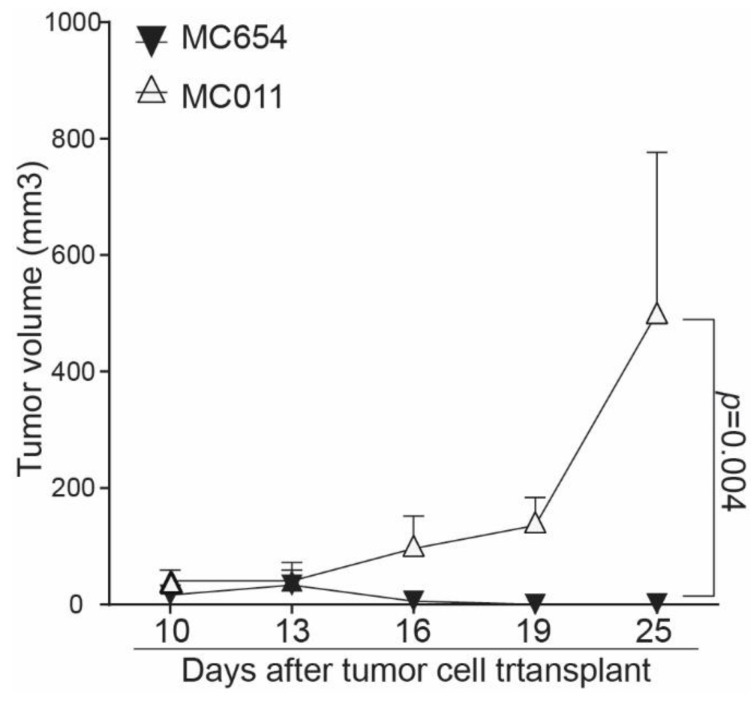
Deletion of Irf8 gene in tumor cells promotes tumor growth in vivo. WT MC654 and IRF8 KO MC011 cells were injected into C57BL/6 mice (5 ×10^5^ cells/mouse). Tumor growth was monitored over time.

**Figure 6 cells-12-00310-f006:**
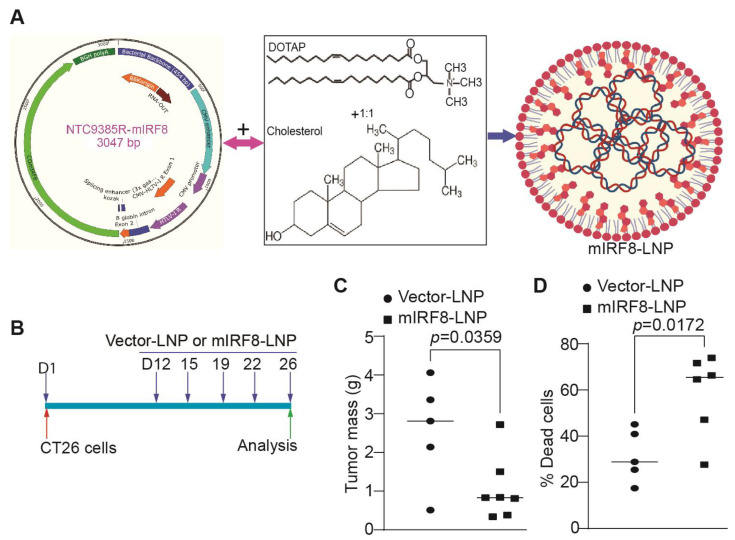
Lipid nanoparticle delivery of IRF8-encoding plasmid DNA to tumor suppresses tumor growth in vivo. (**A**) Formulation of the IRF8-encoding nanoplasmid (NTC9385R-mIRF8) with lipid nanoparticle DOTAP-Cholesterol (1:1 molecular ratio) to produce mIRF8-LNP. (**B**) mIRF8-LNP therapy experimental scheme. CT26 tumor cells were injected into BALB/c mice subcutaneously. The tumor-bearing mice were treated with lipid nanoparticle-encapsulated vector control NTC9385R (Vector-LNP) or NTC9385R-mIRF8 (mIRF8-LNP). (**C**) Tumors were resected at the end of the experiment and measured for size. Shown is quantification of tumor size. (**D**) The tumor tissues were dissected, stained with Zombie UV, and analyzed by flow cytometry to quantify tumor cell death.

**Figure 7 cells-12-00310-f007:**
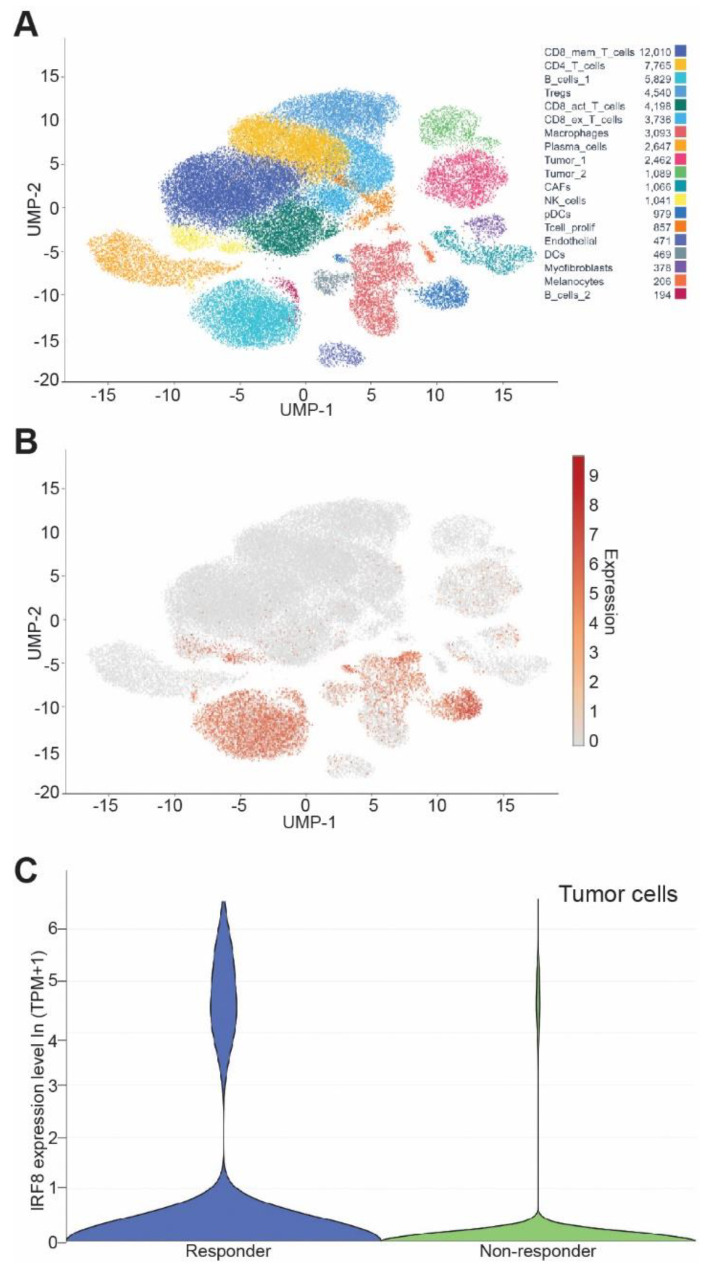
Tumor cell IRF8 expression level correlates with cancer patient response to anti-PD-1 immunotherapy. (**A**) UMAP dimensionality reduction plot of cell types in human melanoma from GEO dataset GSE115978. (**B**) UMAP dimensionality reduction plot of expression level of *Irf8* in tumor resident cells. (**C**) Violin plot of IF8 mRNA level in tumor cells in the responders and non-responders of melanoma patients after nivolumab immunotherapy.

## Data Availability

The RNA-seq dataset will be deposited in the GEO database.

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
