# Peer review of "IRF8 Regulates Intrinsic Ferroptosis through Repressing p53 Expression to Maintain Tumor Cell Sensitivity to Cytotoxic T Lymphocytes"

_cells, 2023, doi:10.3390/cells12020310_

Round 1

Reviewer 1 Report (New Reviewer)

The present manuscript is appropriately designed and executed.

The results could be improved by discussing them appropriately in the discussion part and the images specifically 2B needs to be enlarged.

Author Response

x

Comment 1: The present manuscript is appropriately designed and executed.

Revision 1: Thank you

Comment 2: The results could be improved by discussing them appropriately in the discussion part and the images specifically 2B needs to be enlarged.

Revision 2: We thank the reviewer for this advice. We have revised the discussion and enlarged Fig 2B as advised.

Reviewer 2 Report (New Reviewer)

In this paper, the author described that interferon regulatory factor 8 (IRF8) can inhbits RSL3-induced ferroptosis maybe through inducing the expression of p53. Through analyzing the gene expression of wild type and IRF8 knock-out cells, the author discovered that differential expressed genes induced by IRF8 knockout is enriched in ferroptosis-related genes. And importantly they found that IRF8 expression level maybe an indicator for melanoma patients' response to  immune checkpoint inhibitor immunotherapy.

However, there are many points the author need to address to get this paper published.

(1) there is still missing evidence to support the authors' point that IRF8 regulate ferroptosis through p53.

The author only showed KO p53 make IRF8 KO cells more sensitive to RSL3-induced ferroptosis, but it may also sensitize the WT cells to RSL3-induced ferroptosis. Further evidence are needed to prove the author's point.

(2) Can the author explains why choose MC659/654 and MC011/010. There should be other clones.

(3)In figure.1, ruler is missing in some panel.  And it is better to label the sample name in the figure.

(4)In figure.4, the author should have empty vector/IRF8 wild type and IRF8 K79E together. And can the author show the evidence K79E does not have any IRF8 related activity.

(5) In the context, there are many spelling mistakes and grammatical errors. please carefully check and correct them.

Author Response

Comment 1: there is still missing evidence to support the authors' point that IRF8 regulate ferroptosis through p53.

Revision 1: We thank the reviewer for this comment. This conclusion is made based on the findings: 1) IRF8 KO cells are less sensitive to ferroptosis induction and IRF8 KO resulted in p53 up-regulation in the tumor cells; and 2) Knocking out p53 restored IRF8 KO tumor cell sensitivity to ferroptosis induction. We agree with the reviewer that IRF8 may regulate ferroptosis through other mechanisms, but this manuscript focus on the IRF8-p53 pathway. We have clarified this in discussion section to address the reviewer’s comment.

Comment 2: The author only showed KO p53 make IRF8 KO cells more sensitive to RSL3-induced ferroptosis, but it may also sensitize the WT cells to RSL3-induced ferroptosis. Further evidence are needed to prove the author's point.

Revision 2: The reviewer’s point is well-taken. The reason that we did not test this in WT tumor cells is because p53 is not detectable in WT tumor cells (Fig. 2E). We have clarified this in the result section.

Comment 3: Can the author explains why choose MC659/654 and MC011/010. There should be other clones.

Revision 3: In the literature, gene silencing/knocking out experiments usually use 2 siRNA or 2 CRISPR sgRNAs. In this study, we used 2 WT (MC659 and MC654) and 2 IRF8 KO (MC010 and MC011) tumor cells lines. These 4 cell lines were established from 4 tumor-bearing mice (Fig. 1). Similarly, we used 2 p53 CRISPR KO cell lines (Fig. 3C). Use of 2 pairs of WT and KO cell lines is based on the general practice in the literature. We hope that this is acceptable.

Comment 4: In figure.1, ruler is missing in some panel.  And it is better to label the sample name in the figure.

Revision 4: We thank the reviewer for this comment. DNA ladder/size markers are used in 1A. Sample is labeled with their name is 1A. For 1B-1D, scale bars are used. Panels a-d are used to label the samples since adding sample name to the image would cover the part of images. We have revised the figure legend to clarify this to address this concern.

Comment 5: In figure.4, the author should have empty vector/IRF8 wild type and IRF8 K79E together. And can the author show the evidence K79E does not have any IRF8 related activity.

Revision 5: Figure 4A-B is to compare empty vector/IRF8 wild type vs IRF8 K79E in response to RSSL-3-induced ferroptosis. Figure 4C-D is to compare empty vector/IRF8 wild type vs IRF8 K79E in response to tumor-specific CTLs and Ferr-1 was used as a ferroptosis inhibitor to determine that killing depends on ferroptosis. Therefore, the comparison is between empty/IRF8 wild type vs IRF8 K79E. We have clarified this in the legend. IRF8 K79E plasmid was provided by Dr. Ozato Keiko and was characterized in the literature. We have now pointed this out and referenced the literature (ref 69) of IRF8 functional characterization.   

Comment 6:  In the context, there are many spelling mistakes and grammatical errors. please carefully check and correct them.

Revision 6: We thank the reviewer for pointing this out. We have carefully checked the entire manuscript for errors and made corrections as advised.

Round 2

Reviewer 2 Report (New Reviewer)

The author clearly clarify my concerns, I think this paper is ready to be published.

This manuscript is a resubmission of an earlier submission. The following is a list of the peer review reports and author responses from that submission.

Round 1

Reviewer 1 Report

The quality of the images is poor (Figs.2,3&4). The authors have not made any efforts to improve the quality of the images. I could still see mislabeled figures (e.g. Fig.3A MC011 in the WT group) and Flow dot plots without any gating (Fig.3A).

There are good IRF8 western blotting antibodies available with Cell signaling, R&D, Thermo Fisher, etc. Authors should have tried those WB Abs to strengthen their results.

Cutting a PCR gel image and pasting it with others is not acceptable (Fig.S1). They should have loaded the samples in proper order. 

RSL3 is GPX4 specific inhibitor, if authors conclude that IRF8 significantly decreased the tumor cell sensitivity to ferroptosis induction by RSL3, they should also have validated it by analyzing GPX4 expression after RSL3 treatment to IRF8 KO cells. It would have strengthened the result “p53 blocks RSL3 inhibition of GPX4”. 

Reviewer 2 Report

Thank you for the improved version.

Please check out the sentence on line 64.

Reviewer 3 Report

The authors have addressed the all comments.

Reviewer 4 Report

The manuscript showed the possibility that IRF8 repressed p53 expression to maintain tumor cell sensitivity to intrinsic ferroptosis in CTL-induced tumor cell death pathway. There are several questions that need to be answered.

1. IRF8 appears suddenly in the part of Introduction. Please explain why this molecule was chosen as the research object?

2. IRF8 knockdown promoted tumor growth in vivo, and its overexpression inhibited tumor cell proliferation in vitro. However, IRF8 knockdown could increase the expression of P53 which has been identified a tumor suppressor. More experiments about IRF8-p53 correlation are needed to obtain a clear explanation.

3. Several key indicators related to intrinsic ferroptosis, such as GPX4, ROS and GSH-GSSG levels should be detected.

4. Why repeated the experimental method in Result Two? Also In this result, were HE staining and IHC experiment conditions consistent in Figure 2B and 2D (a-d)?

5. The logic in the part of Introduction is not clear enough. The content should be progressed layer by layer. The discussion part is not in-depth enough to fully reflect the innovation of the experiment.

6. Some sentences were incorrectly expressed. For example, "The cysteine, once taken into the cell, is reduced to cysteine for the synthesis of glutathione (line 65-66)" and "These WT and IRF8 KO tumor cell lines were genotyped as the IRF8 KO mice (line 119)".